# Penalty and Shrinkage Strategies Based on Local Polynomials for Right-Censored Partially Linear Regression

**DOI:** 10.3390/e24121833

**Published:** 2022-12-15

**Authors:** Syed Ejaz Ahmed, Dursun Aydın, Ersin Yılmaz

**Affiliations:** 1Department of Mathematics and Statistics, Brock University, St. Catharines, ON L2S 3A1, Canada; 2Department of Statistics, Mugla Sıtkı Kocman University, 48000 Mugla, Turkey

**Keywords:** local polynomial regression, lasso, elasticnet, SCAD, MCP, partially linear model, right-censored data

## Abstract

This study aims to propose modified semiparametric estimators based on six different penalty and shrinkage strategies for the estimation of a right-censored semiparametric regression model. In this context, the methods used to obtain the estimators are ridge, lasso, adaptive lasso, SCAD, MCP, and elasticnet penalty functions. The most important contribution that distinguishes this article from its peers is that it uses the local polynomial method as a smoothing method. The theoretical estimation procedures for the obtained estimators are explained. In addition, a simulation study is performed to see the behavior of the estimators and make a detailed comparison, and hepatocellular carcinoma data are estimated as a real data example. As a result of the study, the estimators based on adaptive lasso and SCAD were more resistant to censorship and outperformed the other four estimators.

## 1. Introduction

Consider the partially linear (or semiparametric) regression model
(1)$$z_{i}=x_{i}^{\prime }\beta +f\left(t_{i}\right)+\varepsilon_{i},\;i=1,\ldots ,n$$
where $z_{i}{}^{\prime }s$ are the observations of the response variable, $x_{i}=\left(x_{i1},\ldots ,x_{ik}\right)$ is known k− dimensional vectors of explanatory variables, $t_{i}\in \left[a,b\right]$ is the value of an extra explanatory variable t, $\beta ={\left(\beta_{1},\ldots ,\beta_{k}\right)}^{\prime }$ is an unknown *k*-dimensional parameter vector to be estimated, f(.) is an unknown univariate smooth function, and $\varepsilon_{i}{}^{\prime }s$ are supposed to be uncorrelated independent random variables with mean zero and finite variance $\sigma_{\varepsilon }^{2}=E\left(\varepsilon^{2}\right)$. Partially linear models through a nonparametric component are flexible enough to cover many situations; in fact, these models may be an appropriate choice when it is suspected that the response variable z is linearly dependent on x, indicating parametric effects, but nonlinearly related to $t_{i}$ denoting nonparametric effects. Note that model (1) can be expressed in matrix and vector form as
(2)Z=Xβ+f+ε
where $Z={\left(z_{1},\ldots ,z_{n}\right)}^{\prime }$, $X={\left[x_{1},\ldots ,x_{n}\right]}^{\prime }$ is an (n×k) design matrix with $x_{i}={\left(x_{i1},\ldots ,x_{ik}\right)}^{\prime }$ denoting the i.th k−dimensional row vector of X, $f=\left(f\left(t_{1}\right),\ldots ,f\left(t_{n}\right)\right){\;}^{\prime }$, and $\varepsilon =\left(\varepsilon_{1},\varepsilon_{2},\ldots ,\varepsilon_{n}\right)$ is a random error vector with E(ε)=0 and $Var\left(\varepsilon \right)=\sigma^{2}I_{n}$. For more discussions on model (1), see [1,2,3], among others.

In this paper, we are interested in estimating the parametric and nonparametric components of model (1) when the observations of the response variable are incompletely observed and right-censored by the random censoring variable $c_{i}$, i=1, 2, … , n, but $x_{i}$ and $t_{i}$ are completely observed. In the case where $z_{i}$’s are the censored from the right, then any estimation procedure cannot be applied to $z_{i}$ due to censoring. To add the effect of the censorship to the model estimation process, it should be revealed with the help of auxiliary variables that the censorship problem should be solved accordingly. Therefore, instead of observing the values of the response variable $z_{i}$, we observe the dataset ($y_{i},\;\delta_{i}$) with
(3)$$y_{i}=\min\left(z_{i},c_{i}\right),\;\delta_{i}=I\left(y_{i}<c_{i}\right),\;i=1,2,\ldots ,n$$
where $y_{i}$’s are the observations of the updated new response variable according to censorship and $\delta_{i}$’s are the values of the censor indicator related to$\;y_{i}$’s. If the ith observation of $z_{i}$ is censored, we choose $y_{i}=c_{i}$ and $\delta_{i}=0$; otherwise, we consider $y_{i}=z_{i}$ and $\delta_{i}=1.$ In this case, model (1) transforms into a semiparametric model with the right-censored data, which can also be updated in terms of the values of the new response variable.

In the literature, there are several studies about the right-censored linear model (f(t)=0 in model (1)), including [4,5,6,7]. The right-censored nonparametric regression model (β=0 in model (1)) has been studied by [8,9], among others. In addition, right-censored partially linear models have been studied by [10], who used smoothing splines based on Kaplan–Meier weights as an estimation procedure, and [11] considered censored partial linear models and illustrated the theoretical properties of the semiparametric estimators based on the synthetic data transformation. Aydın and Yilmaz [12] suggested three semiparametric estimators based on regression splines, kernel smoothing, and smoothing spline methods using synthetic data transformation. Regarding the partially linear models with penalty functions, in the case of the noncensored data, [13] studied the estimation of the semiparametric model based on the two absolute penalty functions, which are lasso and adaptive lasso with B-splines. Moreover, they conducted a technical analysis of the estimators meticulously with asymptotic properties.

This paper considers model (1) under a right-censored response variable and a large number of covariates in the parametric component. Notice that right-censored data cause biased estimates due to incomplete observations that manipulate the data structure. Therefore, if the censorship is ignored, inferences based on estimated models may be wrong or deviated. For instance, in clinical trials, some of the observed patients may withdraw from the study before it ends, or they may die from another reason, which makes the corresponding observation right-censored. In particular, in medical studies as in the given example, preventing information loss and obtaining less biased estimates are quite important. This paper, therefore, solves both variable selection and censorship problems. To achieve the variable selection, six different penalty functions are considered: ridge, lasso, adaptive lasso, SCAD, MCP, and elasticnet. Notice that a detailed study about penalty functions and shrinkage techniques is provided by [14]. Local polynomial regression is used as the smoothing method. Finally, the censorship problem is solved using the synthetic data transformation proposed by [6].

In the light of the information given, the main difference and most important contribution of this article from previous studies is that it proposes quasi-parametric estimators based on six different penalty functions with the local polynomial technique for the right-censored model (1). To the best of our knowledge, this kind of detailed study has not yet been made in the literature.

The paper is organized as follows: Section 2 introduces the right-censored data phenomenon and some preliminaries. Section 3 explains the local polynomial smoothing method, and the modified semiparametric estimators are introduced based on the six penalty functions with theoretical properties. In Section 4, the evaluation metrics are shown. Section 5 performs a Monte Carlo simulation study, and the results are presented. Section 6 presents an analysis of hepatocellular carcinoma data as a real data example. Finally, conclusions are given in Section 7.

## 2. Preliminaries and General Estimation Procedure

Let H, F, and G be distribution functions of the variables yi, zi and ci, respectively. More precisely, let the probability distribution functions of these variables be
H(u)=P(yi≤u), F(u)=P(zi≤u), and G(u)=P(ci≤u) for u∈ℝ
and their corresponding survival functions are given by
$$\bar{H}\left(u\right)=1-H\left(u\right)=P\left(y_{i}>u\right),\;\;\bar{F}\left(u\right)=P\left(z_{i}>u\right),\;\mathrm{and}\;\bar{G}\left(u\right)=P(c_{i}>u).$$

The key idea here is to examine the effect of the explanatory variables on the response variable by estimating the expected value of E(z|x,t) by the regression function. In the setting of a semiparametric regression problem, first, we need to make some identification conditions on the response variable, censoring, and explanatory variables and their dependence relationships. In other words, we take some assumptions to ensure that the model is identifiable.

**Assumption** **1.**(i)$\;z_{i}\;$*and* $c_{i}$ *are conditionally independent given* $\left(x_{i},\;t_{i}\right)$; (ii) $P\left(\;z_{i}\leq c_{i}|x_{i},\;t_{i},\;z_{i}\right)=P\left(\;z_{i}\leq c_{i}|\;z_{i}\right)$.

It should be emphasized that Assumption 1(i) and Assumption 1(ii) are commonly accepted assumptions regarding right-censored models and survival analysis (see [15,16]). Assumption 1(i) is an independency condition that provides identifiability for the model. Assumption 1(ii) indicates that covariates provide the same information about the response variable independent of the existence of censorship (see [17]).

Because of the censoring, the classical methods for estimating the parametric and nonparametric components of model (3) are inapplicable. The most important reason for this is that the censored observations $z_{i}$ and updated random observations $y_{i}$ have different expectations. This problem can be overcome by using so-called synthetic data, as in censored linear models. We refer, for example, to the studies of [6,12], among others. In this context, when the distribution G is known, we use synthetic data transformation
(4)$$y_{iG}^{*}=\delta_{i}y_{i}{\left\{1-G\left(y_{i}\right)\right\}}^{-1}=\delta_{i}y_{i}{\left\{\bar{G}\left(y_{i}\right)\right\}}^{-1}$$
where $\bar{G}\left(.\right)=1-G\left(.\right)$ and G(.) denotes the distribution functions of censoring variables C, as defined in the introduction to this section. The nature of the synthetic data method ensures that $\left(Y_{iG}^{*},X_{i},\;W_{i}\right),\;i=1,2,\ldots ,n$ are independent random variables with $E\left(y_{iG}^{*}|x_{i},\;t_{i}\right)=E\left(z_{i}|x_{i},\;t_{i}\right)$, as described in Lemma 1.

**Lemma** **1.***If, instead of response observations* $z_{i},\;$*only* ${\left\{\left(y_{i},\delta_{i}\right)\right\}}_{i=1}^{n}$ *is observed in the context of a semiparametric regression model and the censoring distribution* G *is known, then the regression function (or mean vector)* $\mu =x_{i}^{\prime }\beta +f(\beta +f\left(t_{i}\right)$ *is a conditional expectation; that is,*$\;E\left(y_{iG}^{*}|x_{i},\;t_{i}\right)=E\left(z_{i}|x_{i},\;t_{i}\right)=\mu$.

*A proof of Lemma 1 is given in Section A.1*.

Lemma 1 cannot be directly applied to the estimation f(⋅) if the distribution G is unknown. To overcome this difficulty, [6] recommends replacing G with its Kaplan–Meier estimator [18]:(5)$$\hat{G}\left(t\right)=1-\overset{n}{\underset{i=1}{\prod }}{\left(\frac{n-i}{n-i+1}\right)}^{I\left[y_{\left(i\right)}\leq \;t,\;\delta_{\left(i\right)}=0\right]},\;t\geq 0$$
where $y_{\left(1\right)}\leq \cdots \leq y_{\left(n\right)}$ are the order statistics of $y_{1},\ldots ,y_{n}$ and $\delta_{\left(i\right)}$ is the corresponding indicator related to $y_{\left(i\right)}$, as defined in previous sections. In this case, that is, when the distribution G is unknown, we consider the following synthetic data transformation:(6)$$y_{i\hat{G}}^{*}=\delta_{i}y_{i}{\left\{1-\hat{G}\left(y_{i}\right)\right\}}^{-1},\;i=1,2,\ldots ,n$$

## 3. Local Polynomial Estimator

Consider the semiparametric regression model defined in (1). Here, we approximate the regression function $f\left(t_{i}\right)$ locally by a polynomial of order *p* (see [19]). Using a Taylor series expansion in $t_{i}$ at a neighborhood of fixed $t_{i0}$, the p.th degree polynomial approximation of $f\left(t_{i}\right)$ yields
(7)$$f\left(t_{i}\right)\approx \overset{p}{\underset{j=0}{\sum }}\frac{f^{\left(j\right)}\left(t_{i0}\right)}{j!}{\left(t_{i}-t_{i0}\right)}^{j}=\overset{p}{\underset{j=0}{\sum }}b_{j}{\left(t_{i}-t_{i0}\right)}^{j}$$

Note that fixed $t_{i0}$ is determined in the range $t_{i0}\in \left[t_{i}-\delta ,t_{i}+\delta \right]$ for a small real-valued δ and used to estimate $t_{i}$ locally (see [8] for details). The idea is to estimate the components of a semiparametric model, leading to the minimization of the local weighted least squares criterion:(8)$$\underset{b,\beta }{\min}\overset{n}{\underset{i=1}{\sum }}{\left\{y_{i\hat{G}}^{*}-\overset{p}{\underset{j=0}{\sum }}{\left(t_{i}-t_{i0}\right)}^{j}b_{j}-x_{i}^{\prime }\beta \right\}}^{2}K\left(\frac{t_{i}-t_{i0}}{h}\right)\;$$
where $y_{i\hat{G}}^{*}{}^{\prime }s$ are the values of the synthetic variable, as defined in (6), K(.) is a kernel function assigning weights to each point, and h is the bandwidth parameter controlling the size of the local neighborhood of $t_{i0}$. Additionally, note that vector and matrix notation (8) can be written as follows:(9)$$\underset{b,\beta }{\min}{\left(y_{\hat{G}}^{*}-Tb-X\beta \right)}^{\prime }W_{n}\left(\;y_{\hat{G}}^{*}-Tb-X\beta \right)$$
where$\;y={\left(y_{1},\ldots ,y_{n}\right)}^{\prime },\;b={\left(b_{0},\ldots ,b_{p}\right)}^{\prime },\;W_{n}=diag\left(K\left(\frac{t_{i}-t_{i0}}{h}\right)\right)$ is a n×n weights matrix whose properties are provided in Assumption 4. Note that the minimum problem (9) has a unique solution based on the following matrices:$$T=\left[\begin{array}{ccc} 1 & \left(t_{1}-t_{i0}\right) & \begin{array}{cc} \cdots & {\left(t_{1}-t_{i0}\right)}^{p} \end{array}\\ 1 & \left(t_{2}-t_{i0}\right) & \begin{array}{cc} \cdots & {\left(t_{2}-t_{i0}\right)}^{p} \end{array}\\ \begin{array}{c} \vdots \\ 1 \end{array} & \begin{array}{c} \vdots \\ \left(t_{n}-t_{i0}\right) \end{array} & \begin{array}{cc} \begin{array}{c} \vdots \\ \ldots \end{array} & \begin{array}{c} \vdots \\ {\left(t_{n}-t_{i0}\right)}^{p} \end{array} \end{array} \end{array}\right]\;and\;X=\left[\begin{array}{ccc} x_{11} & x_{12} & \begin{array}{cc} \cdots & x_{1k} \end{array}\\ x_{21} & x_{22} & \begin{array}{cc} \ldots & x_{2k} \end{array}\\ \begin{array}{c} \vdots \\ x_{n1} \end{array} & \begin{array}{c} \vdots \\ x_{n2} \end{array} & \begin{array}{c} \begin{array}{cc} \vdots & \;\;\;\;\vdots \end{array}\\ \begin{array}{cc} \ldots & x_{nk} \end{array} \end{array} \end{array}\right]$$

For technical convenience, we assume that β is known to be the true parameter. Then the solution to minimizing (9) is
(10)$$\hat{b}={\left(T^{\prime }W_{n}T\right)}^{-1}T^{\prime }W_{n}\left(y_{\hat{G}}^{*}-X\beta \right)$$

It can be seen from the Taylor series expansion given in (7) that one needs to select the first element of the vector  b^=(b^0,…,b^k) in order to obtain f^(t0)=b^0. Then, for the fixed neighborhood t0, the deconvoluted local polynomial estimator of the regression function can be written as
(11)f^(t0;h)=∑i=1nω1′(ti′Wniti)−1ti′Wni(yiG^*−xi′β)=ω1′(T′WnT)−1T′Wn(y−Xβ)=Sh(yG^*−Xβ)
where Sh=ω1′(T′WnT)−1T′Wn denotes the deconvoluted local polynomial smoother matrix, ω1′=(1, t,…,tp) ∈ ℝ(p+1) dimensional matrix having 1 in the first position and 0 otherwise, and the matrices T and Wn are as defined above.

After the theoretical confirmation by giving Equations (10) and (11), the cases of both model parameters (β,f) are unknown. To obtain the local polynomial-based estimates $\left({\hat{\beta }}_{L},{\hat{f}}_{L}\right)$, the smoother matrix $S_{h}$ given right after Equation (11) is used to calculate the following partial residuals in matrix form:(12)$$\;\tilde{X}=\left(I_{n}-S\right)X=x_{i}-\overset{n}{\underset{i=1}{\sum }}\omega_{i}{}^{\prime }{\left(t_{i}{}^{\prime }W_{ni}t_{i}\right)}^{-1}t_{i}{}^{\prime }W_{ni}x_{i}={\tilde{x}}_{i}$$
and
(13)$${\tilde{y}}_{\hat{G}}^{*}=\left(I_{n}-S\right)y_{\hat{G}}^{*}=y_{i\hat{G}}^{*}{}_{i}-\overset{n}{\underset{i=1}{\sum }}\omega_{i}{}^{\prime }{\left(t_{i}{}^{\prime }W_{ni}t_{i}\right)}^{-1}t_{i}{}^{\prime }W_{ni}y_{i\hat{G}}^{*}={\tilde{y}}_{i\hat{G}}^{*}$$
where
$$S=\left(\begin{array}{c} {\left(S_{h}\right)}_{t_{1}}\\ \vdots \\ {\left(S_{h}\right)}_{t_{n}} \end{array}\right)=\left(\begin{array}{c} \left(1\;0\ldots .0\right){\left(T_{t_{1}}^{\prime }W_{t_{1}}T_{t_{1}}\right)}^{-1}T_{t_{1}}{}^{\prime }W_{t_{1}}\\ \vdots \\ \left(1\;0\ldots .0\right){\left(T_{t_{n}}^{\prime }W_{t_{n}}T_{t_{n}}\right)}^{-1}T_{t_{n}}{}^{\prime }W_{t_{n}} \end{array}\right)$$

Thus, we obtain a transformed set of data based on local residuals. Considering these partial residuals for the vector β yields the following least squares instead of criterion (9):(14)$$\underset{\beta }{\min}\overset{n}{\underset{i=1}{\sum }}{\left\{{\tilde{y}}_{i\hat{G}}^{*}-{\tilde{x}}_{i}^{\prime }\beta \right\}}^{2}={\left||({\tilde{y}}_{\hat{G}}^{*}-\tilde{X}\beta )|\right|}^{2}$$
where ${\tilde{x}}_{i}^{\prime }$ is the i.th row of the matrix $\tilde{X}$. Under Assumptions 2–4, by applying the least squares technique to (14), we obtain a *“modified local polynomial estimator”* ${\hat{\beta }}_{L}\;$for the parametric part of the semiparametric model (3), given by
(15)$${\hat{\beta }}_{L}={\left({\tilde{X}}^{\prime }\tilde{X}\right)}^{-1}{\tilde{X}}^{\prime }{\tilde{y}}_{\hat{G}}^{*}$$

Correspondingly, a “modified local polynomial estimator” ${\hat{f}}_{L}\;$of the function f(.) for the nonparametric part in the semiparametric model (3) is defined as
(16)$${\hat{f}}_{L}=S\left(y_{\hat{G}}^{*}-X{\hat{\beta }}_{L}\right)$$

The implementation details of the Equations (15) and (16) are given in Appendix A2. We conclude this section with the following assumptions necessary to obtain the main results. These assumptions are quite general and easily fulfilled.

**Assumption** **2.***When the covariates (*$x_{ij},t_{i}$*) are fixed design points, there exist continuous functions* $h_{j}\left(.\right)$ *defined on* [0,1] *such that each component of* $x_{i}$ *satisfies*$$x_{ij}=h_{j}\left(t_{i}\right)+v_{ij}\;,1\leq i\leq n,\;1\leq j\leq k$$*where* $\left\{v_{ij}\;\right\}$ *is a sequence of real numbers satisfying*$$\underset{n\to \infty }{\lim}\frac{1}{n}\sum_{i=1}^{n}v_{is}v_{im}^{\prime }=c_{sm},1\leq s\leq k,\;1\leq m\leq k$$*and* $C=\left(c_{sm}\right)$ *is a (*k×k*) dimensional nonsingular matrix.*

**Assumption** **3.***The functions* f(.) *and* $h_{j}\left(.\right)$ *are Lipschitz continuous of order 1 for* =1,…,k.

Note that Assumption 2 generalizes the conditions of [20,21], where ($x_{ij},t_{i}$) are fixed design points for a partially linear model with uncensored data. Assumption 3 is required to establish asymptotic normality with an observed value $t_{i}.$

**Assumption** **4.***The weight functions* $W_{n}$ *satisfy these conditions:*(i.)$max_{1\leq i\leq n}\overset{n}{\underset{j=1}{\sum }}W_{ni}\left(t_{j}\right)=O\left(1\right)$(ii.)$max_{1\leq i,j\leq n}\overset{n}{\underset{j=1}{\sum }}W_{ni}\left(t_{j}\right)=O\left(n^{-2/3}\right)$(iii.)$max_{1\leq i\leq n}\overset{n}{\underset{j=1}{\sum }}W_{ni}\left(t_{j}\right)I\left(\left|t_{i}-t_{j}\right|>a_{n}\right)=O\left(b_{n}\right),\;$*where* I(.) *is an indicator function,* $a_{n}$ *satisfies* $lim\;sup_{n\to \infty }na_{n}^{3}<\infty$*, and* $b_{n}$ *satisfies* $lim\;sup_{n\to \infty }nb_{n}^{3}<\infty$.

### 3.1. Ridge-Type Local Polynomial Estimator

In this paper, we confine ourselves to the local polynomial estimators of the vector parameter β and the unknown smooth function f(.) in a semiparametric model. For a given bandwidth parameter h, the corresponding estimators β and f based on model (2) are described by (14) and (15), respectively. Multiplying both sides of model (2) by $\left(I_{n}-S\right)$, we obtain
(17)$$\tilde{Z}=\tilde{X}\beta +\tilde{\varepsilon }$$
where $\tilde{Z}=\left(I_{n}-S\right)Z$, $\tilde{\varepsilon }=\tilde{f}+\varepsilon^{*},$ $\tilde{f}=\left(I_{n}-S\right)f$, and $\varepsilon^{*}=\left(I_{n}-S\right)\varepsilon$, similar to (12) and (13).

This consideration turns model (17) into an optimization problem to obtain the estimator of the vector β corresponding parametric part of the semiparametric model in (2). In this context, this model leads to the following penalized least squares (PLS) criterion for the ridge regression problem:(18)$$PLS_{RL}=\underset{\beta }{arg\;min}{\left(\tilde{Z}-\tilde{X}\beta \right)}^{\prime }\left(\tilde{Z}-\tilde{X}\beta \right)+\lambda \beta^{\prime }\beta$$
where λ is a positive shrinkage parameter that controls the magnitude of the penalty. The solution to this minimization problem (17) provides the following Theorem 1.

**Theorem** **1.***Ridge-type local polynomial estimator for* β *is presented by* ${\hat{\beta }}_{RL}\;$*and is expressed based on the local polynomial smoothing matrix* S *by*(19a)$${\hat{\beta }}_{RL}\left(\lambda \right)={\left({\tilde{X}}^{\prime }\tilde{X}+\lambda I_{k}\right)}^{1}{\tilde{X}}^{\prime }{\tilde{y}}_{\hat{G}}^{*}$$*where* ${\tilde{y}}_{\hat{G}}^{*}$ *is a vector of updated response observations, as defined in Equations (6) and (13).*

*A proof of Theorem 1 is given in Section A.3*.

As shown in Theorem 1, when λ=0, the ridge-type local polynomial estimate reduces to an ordinary least squares estimate problem based on the local residuals defined in Equations (12) and (13). It should be noted that in order to estimate the unknown function f, we imitate Equation (16) and define
(19b)$${\hat{f}}_{RL}=S\left(y_{\hat{G}}^{*}-X{\hat{\beta }}_{RL}\left(\lambda \right)\right)$$

Thus, the estimator (19b) is stated as the ridge-type local polynomial estimator of the unknown function f in the semiparametric model (1.2).

### 3.2. Penalty Estimation Strategies Based on Local Polynomial

Several penalty functions are discussed for linear and generalized regression models in the literature (see [22]). In this paper, we study the minimax concave penalty (MCP), the least absolute shrinkage and selection operation (lasso), the smoothly clipped absolute deviation method (SCAD), the adaptive lasso, and the elasticnet method, which is a regularized regression technique that linearly combines the $L_{1}$ and $L_{2}$ penalties of the lasso and the ridge regression methods, respectively.

In this paper, we suggest local polynomial estimators based on different penalties for the components of the semiparametric regression model. For a given penalty function $P_{\lambda }\left(\beta \right)$ and tuning parameter λ, the general form of the penalized least squares ($PLS_{G}$) of penalty estimators can be expressed as
(20)$$PLS_{G}=\underset{\beta }{arg\;min}\left\{\overset{n}{\underset{i=1}{\sum }}{\left({\tilde{y}}_{i\hat{G}}^{*}-{\tilde{x}}_{i}^{\prime }\beta \right)}^{2}\right\}+P_{\lambda }\left(\beta \right)={\left({\tilde{y}}_{\hat{G}}^{*}-\tilde{X}\beta \right)}^{\prime }\left({\tilde{y}}_{\hat{G}}^{*}-\tilde{X}\beta \right)+P_{\lambda }\left(\beta \right)$$

Note that the vector $\hat{\beta }$ that minimizes (20) for lasso and ridge penalties is known as a bridge estimator, proposed by [23]. On the other hand, elasticnet, SCAD, MCP, and adaptive lasso involve different penalties, which are inspected in the remainder of this paper. It should be emphasized that in the mentioned four penalty functions, in the penalty term $P_{\lambda }\left(\beta \right)=\lambda \sum^{\;}||\beta_{j}||{}_{q}^{q}\;$satisfies the $L_{q}$ norm of the regression coefficients $\beta_{j}$ (see [24,25,26]). Thus, the different penalty estimators corresponding to the parametric and nonparametric components of the semiparametric model can be defined for different values of degree q and shrinkage parameter λ.

#### 3.2.1. Estimation Procedure for the Parametric Component

From (20), we see that for q=2, ridge estimates corresponding to the parametric component can be obtained by minimizing the following penalized residual sum of squares
(21)$${\hat{\beta }}_{RL}=\underset{\beta }{arg\;min}\left\{\overset{n}{\underset{i=1}{\sum }}{\left({\tilde{y}}_{i\hat{G}}^{*}-{\tilde{x}}_{i}^{\prime }\beta \right)}^{2}+\lambda \overset{k}{\underset{j=1}{\sum }}{\left|\beta_{j}\right|}^{2}\right\}$$
where ${\tilde{y}}_{i\hat{G}}^{*}$ is the *ith* synthetic observation of ${\tilde{y}}_{\hat{G}}^{*}\;$and ${\tilde{x}}_{i}^{\prime }$ is the i.th row of the matrix $\tilde{X}$. Notice that the solution (21) has the same regularization estimate stated in (19a). It should also be noted that when q=1 in (20), we obtain the estimator known as the lasso.

**Lasso:** Proposed by [24], lasso, a penalized least squares method, is a regularization method for simultaneous estimation and variable selection that estimates with the $L_{1}$ penalty. The modified local polynomial estimators based on the lasso penalty can be defined as
(22)$${\hat{\beta }}_{LL}=\underset{\beta }{arg\;min}\left\{\overset{n}{\underset{i=1}{\sum }}{\left({\tilde{y}}_{i\hat{G}}^{*}-{\tilde{x}}_{i}^{\prime }\beta \right)}^{2}+\lambda \overset{k}{\underset{j=1}{\sum }}\left|\beta_{j}\right|\right\}$$

Although Equation (22) may seem subtle, the absolute penalty term makes it impossible to find an analytical solution for the lasso. Initially, lasso solutions are obtained through quadratic programming.

**Adaptive lasso:** Zou [25] suggested modifying the lasso penalty by using adaptive weights on $L_{1}$ penalties on the regression coefficients. This weighted lasso, which has oracle properties, is referred to as the adaptive lasso. The local polynomial estimator ${\hat{\beta }}_{aLL}\;$using the adaptive lasso penalty can be defined as follows:(23)$${\hat{\beta }}_{aLL}=\underset{\beta }{arg\;min}\left\{\overset{n}{\underset{i=1}{\sum }}{\left({\tilde{y}}_{i\hat{G}}^{*}-{\tilde{x}}_{i}^{\prime }\beta \right)}^{2}+\lambda \overset{k}{\underset{j=1}{\sum }}{\hat{w}}_{j}\left|\beta_{j}\right|\right\}$$
where w is a weight function given by
$${\hat{w}}_{j}=\frac{1}{{\left|{\hat{\beta }}^{*}\right|}^{q}},q>0$$

It should be noted that ${\hat{\beta }}^{*}$ is an appropriate estimator of β here. For example, an ordinary least squares (OLS) estimate can be used as a reference value. To obtain the adaptive lasso estimates in (23), it is necessary to choose q>0 and compute the weights after obtaining the OLS estimate.

**SCAD:** A disadvantage of the lasso method is that the penalty term is linear in the size of the regression coefficient, so it tends to give highly biased estimates for large regression coefficients. To account for this bias, [26] proposed a SCAD penalty obtained by replacing $\left|\beta_{j}\right|$ in (22) with $P_{\alpha ,\lambda }\left|\beta_{j}\right|$. A modified local estimator ${\hat{\beta }}_{SL}$ based on the SCAD penalty can be described as
(24a)$${\hat{\beta }}_{SL}=\underset{\beta }{arg\;min}\left\{\overset{n}{\underset{i=1}{\sum }}{\left({\tilde{y}}_{i\hat{G}}^{*}-{\tilde{x}}_{i}^{\prime }\beta \right)}^{2}+\overset{k}{\underset{j=1}{\sum }}P_{\alpha ,\lambda }\left|\beta_{j}\right|\right\}$$
where $P_{\alpha ,\lambda }\left(.\right)$ is the SCAD penalty defined by
(24b)$$P_{\alpha ,\lambda }=\lambda \left\{I\left(\left|\beta \right|\right)\leq \lambda +\frac{{\left(\alpha \lambda -\left|\beta \right|\right)}_{+}}{\left(\alpha -1\right)\lambda }I(\left|\beta \right|>\lambda )\right\},\;\mathrm{and}\;\mathrm{for}\;\lambda \geq 0\;$$

It should be stated that here λ>0 and α>2 are the penalty parameters, I(.) is the indicator function, and ${\left(t\right)}_{+}=\max\left(t,0\right)$. In addition, (24b) is equivalent to the $L_{1}$ penalty for α=∞.

**Elasticnet:** The elastic net, proposed by [27], is a penalized least squares regression technique that has been widely used in regularization and automatic variable selection to select groups of correlated variables. Note that the elastic net method linearly combines the $L_{1}$ penalty term, which enforces the sparsity of the elastic net estimator, and the $L_{2}$ penalty term, which ensures appropriate selection of correlated variable groups. Accordingly, the modified local estimator ${\hat{\beta }}_{ENL}$ using an elasticnet penalty is the solution of the following minimization problem:(25)$${\hat{\beta }}_{ENL}=\underset{\beta }{arg\;min}\left\{\overset{n}{\underset{i=1}{\sum }}{\left({\tilde{y}}_{i\hat{G}}^{*}-{\tilde{x}}_{i}^{\prime }\beta \right)}^{2}+\lambda_{1}\overset{k}{\underset{j=1}{\sum }}{\left|\beta_{j}\right|}^{2}+\lambda_{2}\overset{k}{\underset{j=1}{\sum }}\left|\beta_{j}\right|\right\}$$
where $\lambda_{1}$ and $\lambda_{2}$ are the postive regularization parameters. Equation (25) ensures the estimates corresponding to the parametric part of the semiparametric regression model (2), as in the other methods.

**MCP:** Introduced by [28], MCP is an alternative method to obtain less biased estimates of the nonzero regression coefficients in a sparse model. For the given regularization parameters λ>0 and α>0, the local polynomial estimator ${\hat{\beta }}_{MCL}$ based on the MCP penalty can be defined as
(26)$${\hat{\beta }}_{MCL}=\underset{\beta }{arg\;min}\left\{\overset{n}{\underset{i=1}{\sum }}{\left({\tilde{y}}_{i\hat{G}}^{*}-{\tilde{x}}_{i}^{\prime }\beta \right)}^{2}+\overset{k}{\underset{j=1}{\sum }}P_{\alpha ,\;\lambda }\left(\left|\beta_{j}\right|\right)\right\}$$
where $P_{\alpha ,\;\lambda }\left(.\right)$ is the MCP penalty given by
$$P_{\alpha ,\;\lambda }\left(\beta \right)=\int_{0}^{\left|\beta \right|}{\left(\frac{\lambda -x}{\alpha }\right)}_{+}dx=\left(\lambda \left|\beta \right|-\frac{\beta^{2}}{2\alpha }\right)I\left(0\leq \left|\beta \right|<\lambda \alpha \right)+\frac{\lambda^{2}\alpha }{2}I\left(\left|\beta \right|\geq \lambda \alpha \right)$$

#### 3.2.2. The Estimation Procedure for the Nonparametric Component

Equations (21)–(26) provide modified local polynomial estimates based on different penalties for the parametric part of the semiparametric model in (2). Similar in spirit to (19b), the vector of estimated parametric coefficients ${\hat{\beta }}_{LL}$ given in (21) can be used to construct the estimation of the nonparametric part in the same model. In this case, we obtain the modified local estimates based on the lasso penalty of the unknown function, given by
(27)$${\hat{f}}_{LL}=S\left(y_{\hat{G}}^{*}-X{\hat{\beta }}_{LL}\right)$$
as defined in the previous section.

Note that when ${\hat{\beta }}_{aLL}$ defined in (23) is written instead of ${\hat{\beta }}_{LL}$ in Equation (27), local estimates of the nonparametric part based on the adaptive lasso penalty are obtained and are stated as ${\hat{f}}_{aLL}$ symbolically. Similarly, replacing ${\hat{\beta }}_{LL}$ in (27) with ${\hat{\beta }}_{SL},$ ${\hat{\beta }}_{ENL}$, and ${\hat{\beta }}_{MCL}$ yields modified local polynomial estimators, denoted as ${\hat{f}}_{SL}$, ${\hat{f}}_{ENL},\;$and ${\hat{f}}_{MCL}\;$based on the SCAD, elasticnet, and MCP penalties, respectively, for the nonparametric part of the right-censored semiparametric model (2).

#### 3.2.3. Some Remarks on the Penalties

Remarks on the penalties can be stated as follows:
The regularization based on the $L_{1}$ norm produces sparse solutions as well as feature selection. However, the $L_{2}$ norm produces nonsparse solutions and does not have feature selection.Although all of the regularization methods shrink most of the coefficients towards zero, SCAD, MCP, and adaptive lasso apply less shrinkage to nonzero coefficients. This is known as bias reduction.As noted earlier, the tuning parameter α, used for SCAD and MCP estimations, controls how quickly the penalty rate goes to zero. This affects the bias and stability of the estimates, in the sense that there is a greater chance for more than one local minimum to exist as the penalty becomes more concave.As α→∞, the MCP and SCAD penalties converge to the $L_{1}$ norm penalty. Conversely, as α→0, the bias is minimized, but both MCP and SCAD estimates become unstable. Note also that lower values of the tuning parameter α for SCAD and MCP produce more highly variable, but less biased, estimates.The elasticnet penalty is designed to deal with highly correlated covariates more intelligently than other sparse penalties, such as the lasso. Note that the lasso penalty tends to choose one among highly correlated variables, while elasticnet uses them all.


## 4. Performance Indicators

Several performance measurements are described in this section with which to evaluate the performance of the modified six semiparametric local polynomial estimators based on penalty functions: ridge (RL), lasso (LL), adaptive lasso (aLL), SCAD (SL), MCP (MCL), and elasticnet (ENL). Note that the abbreviations given in parentheses here denote the estimators. The performance of the estimators are examined individually for the parametric component, nonparametric component, and overall estimated model. Accordingly, evaluation metrics are given by:

### 4.1. Measures for the Parametric Component

**Root mean squared error (**RMSE**)** of estimated regression coefficients $\left(\hat{\beta }\right)$. The calculation of RMSE is given by:(28)$$RMSE\left(\beta ,\hat{\beta }\right)=\sqrt{k^{-1}{\left(\beta -\hat{\beta }\right)}^{T}\left(\beta -\hat{\beta }\right)}$$
where $\hat{\beta }$ is the obtained estimate of β by any of the introduced six methods. It is replaced by ${\hat{\beta }}_{RL},\;{\hat{\beta }}_{LL},\;{\hat{\beta }}_{aLL},\;{\hat{\beta }}_{SL},$ ${\hat{\beta }}_{ENL}$, and ${\hat{\beta }}_{MCL}$ to obtain the RMSE score for each estimator.

**Coefficient of determination**$\left(R^{2}\right)$ for the estimated models. Note that $R^{2}$ allows us to see overall model performance of the six methods. It can be calculated as follows:(29)$$R^{2}=1-\frac{\sum_{i=1}^{n}{\left(y_{i}-{\hat{y}}_{i}\right)}^{2}}{\sum_{i=1}^{n}{\left(y_{i}-{\bar{y}}_{i}\right)}^{2}},\;i=1,\ldots ,n$$

**Sensitivity, specificity, and accuracy** scores obtained from a confusion matrix. If true values of an interested variable are available, the confusion matrix can be obtained as in Table 1. This matrix allows us to measure the performance of the penalty functions for right-censored data. Accordingly, sensitivity, specificity, and accuracy values can be calculated as follows:(30)acc=(a+d)/(a+b+c+d);sens=a/(a+b);spec=d/(c+d)

**G score** calculated by geometric mean of sensitivity and specificity given in Equation (31):(31)$$G=\sqrt{sens\times spec}$$

### 4.2. Measures for the Nonparametric Component

**Mean squared error (MSE)** is used to measure the performance of the estimated nonparametric components by six methods: ${\hat{f}}_{RL},\;{\hat{f}}_{LL},\;{\hat{f}}_{aLL},\;{\hat{f}}_{SL},$ ${\hat{f}}_{ENL}$, and ${\hat{f}}_{MCL}$**.** Assume that $\hat{f}$ is the fitted nonparametric function obtained from any of the six methods. Accordingly, the MSE is computed as follows:(32)$$MSE\left(\hat{f}\right)=n^{-1}{\left(\overset{n}{\underset{i=1}{\sum }}f\left(t_{i}\right)-\hat{f}\left(t_{i}\right)\right)}^{2}=n^{-1}{\left(f-\hat{f}\right)}^{T}\left(f-\hat{f}\right)$$

**Relative MSE (ReMSE)** is used to make a comparison between performances of the six methods on the estimation of the nonparametric component. The calculation of the ReMSE is given by
(33)$$ReMSE\left({\hat{f}}_{i}\right)=n_{m}^{-1}\left[\underset{i\neq j}{\sum }MSE\left({\hat{f}}_{i}\right)/MSE\left({\hat{f}}_{j}\right)\right]$$
where $n_{m}$ denotes the number of methods, which are six for this paper.

## 5. Simulation Study

We carried out an intense simulation study to evaluate the finite sample performance of the introduced six semiparametric estimators for a right-censored partially linear model. These estimators are compared with each other to evaluate their respective strengths and weaknesses in handling the right-censored problem. *To obtain reproducibility, simulation codes with functions are provided in the following GitHub link:* https://github.com/yilmazersin13?tab=repositories accessed on 31 August 2022. The estimators are computed using the formulations in Section 3. The simulation design and data generation are described as follows:

***Simulation Design***: Two main scenarios are considered for generating the zero and nonzero coefficients of the model because the focus of the penalty functions is on making an accurate variable selection. In each scenario, simulation runs are made for
Three sample sizes: n=50, 150, 300Two censoring levels: CL=10%, 30%Two numbers of parametric covariates k=1540

All possible simulation configurations are repeated 1000 times. To evaluate the performance of the methods, the performance indicators described in Section 4 are used. The scenarios are defined in the data generation section below.

***Data Generation***: Regarding model (1), $z_{i}=x_{i}^{\prime }\beta +f\left(t_{i}\right)+\varepsilon_{i},\;1\leq i\leq n$, each element of the model obtained as
$$x_{i}\sim MN\left[\mu_{k\times 1},\Sigma_{k\times k}\right];\;t_{i}=2.4\left(i-0.5\right)/n,\;f\left(t_{i}\right)=-t_{i}\sin\left(-t_{i}^{2}\right)$$

The true values of regression coefficients are determined for both Scenarios 1 and 2 as follows:$$(\mathrm{Scenario}\;1)\;\;\;\;\;\;\;\beta_{j}=\{\begin{array}{cc} 5 & if\;j=1,\ldots ,5\;\;\;\;\;\;\;\;\;\;\;\;\;\\ -3 & if\;j=11,\ldots ,15\;\;\;\;\;\;\;\;\;\\ 0 & otherwise\;\;\;\;\;\;\;\; \end{array}$$
$$(\mathrm{Scenario}\;2)\;\;\;\;\;\;\;\beta_{j}=\{\begin{array}{cc} 1 & if\;j=1,\ldots ,5\;\;\;\;\;\;\;\;\;\;\;\;\;\;\;\;\\ -0.5 & if\;j=11,\ldots ,15\\ 0 & otherwise \end{array}$$

For both scenarios, there are 10 nonzero $\beta_{j}$’s to be estimated and (k−10) sparse coefficients. The main purpose of using these two scenarios is that it allows us to measure the capacity of the estimators on the selection of nonzero coefficients when $\beta_{j}$’s are close to zero. In addition, these scenarios make it possible to see how the censoring level (CL) affects their performances. *These scenarios allow us to inspect the convergence of the estimated coefficients to the true ones when the sample size is becoming larger practically under censored data, which can be counted as an important contribution of this paper.*

Regarding the censoring data, the censoring variable $c_{i}$ is generated as $c_{i}\sim N\left(\mu_{y},\sigma_{y}^{2}\right)$ independently of the initially observed variable $y_{i}$. An algorithm is provided by [29]. Another important point for this study is the selection of the shrinkage parameters and the bandwidth parameter for the local polynomial approach for the introduced six estimators. In this study, the improved Akaike information criterion $\left(AIC_{c}\right)$, proposed by [30], is used. It can be calculated as follows:$$AIC_{c}\left(\lambda ;h\right)=\log\left(\sigma^{2}\right)+1+\left(\frac{2\left(\sum_{r<k}I\left({\hat{\beta }}_{r}\left(\lambda ,h\right)\neq 0\right)+1\right)}{\left(n-\sum_{r<k}I\left({\hat{\beta }}_{r}\left(\left(\lambda ,h\right)\right)\neq 0\right)-2\right)}\right)$$
where ${\hat{\beta }}_{r}\left(\lambda ,h\right)$ is the estimated coefficient based on the shrinkage parameter λ>0, and the bandwidth h>0 and $\sigma^{2}$ is the variance of the model, and $\underset{r<k}{\sum }I\left({\hat{\beta }}_{r}\left(\lambda ,h\right)\neq 0\right)$ denotes the number of nonzero regression coefficients. Note that, due to the projection (hat) matrix, the introduced estimation procedures (except ridge regression) cannot be written; the number of nonzero coefficients is used instead of the hat matrix.

The results of the simulation study are presented individually for parametric and nonparametric components below. Before that, Figure 1 is presented to provide some information about the generated data. Figure 1 is formed by four panels. Panels (a) and (b) show the original, right-censored, and synthetic response values for CL=10%. Panels (c) and (d) show the same for CL = 30%.

In Figure 1, two plots (i−ii) are represented for the different configurations of Scenarios 1 and 2. In plot (i), scatter plots of the data points for n=50, CL=10%, and k=15 are given. In panel (b) of (i), the working procedure of synthetic data transformation can be seen clearly. It gives zero to right-censored observations and increases the magnitude of observed data points. Thus, it makes equal the expected value of the synthetic response variable and original response variable, as indicated in Section 2. Similarly, plot (ii) shows the scatter plots of data points for n=150, CL=30%, and k=40, which makes it possible to see heavily censored cases. In panel (b) of (ii), due to heavy censorship, the magnitude of the increments in the observed data points is larger than (*i*), which is the working principle of the synthetic data. This is a disadvantage because it significantly manipulates the data structure, although it still solves the censorship problem.

To describe the generated dataset further, Figure 2 shows the nonparametric component of the right-censored semiparametric model. In panel (a), the smooth function can be seen for the small sample size (n=50), low censoring level (CL=10%), and low number of covariates (k=15). Panel (b) shows the nonparametric smooth function for n=150, k=15, and CL=10%. It should be emphasized that the censoring level or number of the parametric covariates does not affect the shape of the nonparametric component. Thus, there is no need to show all possible generated functions here. Note that the nonparametric component affects the number of covariates and the censoring level indirectly in the estimation process.

As previously mentioned, this paper introduces six modified estimators based on penalty and shrinkage strategies. In Figure 3, the shrinkage process of the estimators, according to the shrinkage parameter “lambda,” is provided in panels (i) and (ii). In panel (i), shrunk regression coefficients are shown for scenario 1, n=150, CL=30%, and k=40. Panel (ii) is drawn for scenario 2, n=50, CL=10%, and k=15. When Figure 3 is inspected carefully, it can be seen that in panel (i), due to a high censoring level and many covariates, the shrinkage of the coefficients is more challenging than in panel (ii). One of the reasons for that is, in Scenario 2, coefficients are determined as smaller than the coefficients in Scenario 1 while generating data. For both panels, it can be observed that the SCAD and MCP methods behave similarly. As expected, they shrunk the coefficients quicker than the others. Additionally, lasso and ElasticNet seem close to each other for both panels. However, adaptive lasso differs from the others in both panels. The reason for this is discussed with the results given in Section 5.1.

### 5.1. Analysis of Parametric Component

In this section, the estimation of the parametric component of a right-censored semiparametric model is analyzed, and results are presented for all simulation scenarios in Table 2, Table 3, Table 4 and Table 5 and Figure 4, Figure 5, Figure 6 and Figure 7. The performance of the estimators is evaluated using the metrics given in Section 4: RMSE, $R^{2}$, sensitivity, specificity, accuracy, and G-score. In addition to the performance criteria, the selection ratio of the methods is calculated for the estimators. The selection ratio can be defined as follows:

**Selection Ratio**: The ratio of the selected true nonzero coefficients by the corresponding estimator in 1000 simulation repetitions. The formulation can be given by:$$Selection\;Ratio\left({\hat{\beta }}_{j}\right)=\frac{1}{1000}\overset{1000}{\underset{i=1}{\sum }}I\left({\hat{\beta }}_{j}^{\left(i\right)}\neq 0\right),$$
where ${\hat{\beta }}_{j}^{\left(i\right)}$ is the estimated coefficient for i.th simulation by any of the introduced estimators. Results are given in Figure 6 and Figure 7.

Table 1 and Table 2 include the RMSE scores for the estimated coefficients calculated from (4.1) and $R^{2}$ of the model for Scenarios 1 and 2. The best scores are indicated with bold text. If the two tables are inspected carefully, two observations can be made about the performance of the methods for both Scenarios 1 and 2. Regarding Scenario 1, when the sample size is small (n=50), ENL and SL estimators give smaller RMSE scores and higher $R^{2}$ values than the other four methods. On the other hand, when the sample size becomes larger (n=150, n=300), aLL takes the lead in terms of estimation performance. The results for different censoring levels show that aLL is less affected by censorship than SL and the other methods. This can be observed in Table 1 clearly.

In Table 2, RMSE and $R^{2}$ scores are provided for all simulation configurations of Scenario 2. The results can be distinguished from the results in Table 1 by the higher $R^{2}$ values obtained from the modified ridge estimator. However, the RMSE scores of the modified ridge estimator are the largest. This can be explained by the fact that the ridge penalty uses all covariates, whether sparse or nonsparse. Therefore, the estimated model based on the ridge penalty has larger $R^{2}$ values. On the other hand, the RMSE scores prove that for small sample sizes (n=50), ENL and aLL perform satisfactorily. Moreover, as in the case of Scenario 1, when the sample size becomes larger, aLL gives the most satisfying performance. SL- and LL-based estimators also show good performances in the general frame. If Table 2 is inspected in detail, it can be seen that when k=15, the SL method comes to the front for both low (CL=10%) and high (CL=30%) censoring levels. The same is true for the aLL method regarding strength against censorship.

Figure 4 presents the line plots of the RMSE scores for all simulation cases. As expected, the negative effects of increment on the censoring level and the positive effect of growth on the sample size can be clearly observed from panels (a) and (b). For both scenarios, a peak can be seen when the sample size is small (n=50) and the censorship level increases from 10% to 30%. The methods most affected by censorship are MCL, RL, and SL. The least affected are aLL, lasso, and ENL. Thus, Figure 4 supports the results and inferences obtained from Table 2 and Table 3.

Together with the RMSE scores, one other important metric to evaluate the performance of the parametric component estimation is G-score, which measures the true selection made by the estimators for the sparse and nonzero subsets based on the confusion matrix given in Table 1. In this context, Figure 5 is drawn to illustrate the G-scores of the methods for all simulation combinations using line plots. Note that the G-score changes between the range [0,1] and the lines of methods that are close to 1 are notated as better than the others in terms of successful determination of sparsity.

Figure 5 is formed by two panels: Scenario 1 (left) and Scenario 2 (right). As expected, for all methods except for RL (which does not involve any sparse subset and is therefore not shown in Figure 5 and Table 4), the G-scores diminish when the censoring level is high, and the number of covariates (k) is large. In addition, there is an increasing trend from the small to large sample sizes for LL, aLL, SL, and MCL. This trend is most evident for the aLL line, which makes aLL distinguishable. However, interestingly, ENL is not influenced by the change in sample size, and the G-scores of ENL do not take a value greater than 0.5. In general, aLL, SL, and LL provide the highest G-scores. All G-scores for the simulation study are provided in Table 4 and Table 5 together with the accuracy values of the methods.

Table 4 and Table 5 present the accuracy rates and G-scores for all methods and simulation configurations for both Scenarios 1 and 2. Note that, because the ridge penalty is unable to shrink the estimated coefficients towards zero, the specificity of RL is always calculated as zero. Thus, RL does not have a G-score. When the tables are examined, it can be clearly seen that the prominent methods are aLL, LL, and SL. The aLL produces satisfactory results for each simulation configuration. On the other hand, the other two methods, LL and SL, give good results under different conditions. When this situation is examined in detail, it can be seen that the SL method produces better results when k=15, and the LL method when k=40 with aLL. In addition, it can be said that the level of censorship and the sample size do not affect this situation, except for an increase or decrease in the values. These inferences apply to both scenarios. Here, the difference between the scenarios emerges in the size of the G-scores and accuracy values obtained. It can be said that they are slightly less than the values obtained for Scenario 2.

Unlike the evaluation criteria given in Section 4, the frequency of choosing the correct coefficients for each method in the simulation study is analyzed, and the results are presented for both scenarios in Figure 6 and Figure 7 with bar plots. Figure 6 presents two panels (I and ii), which demonstrate the impact of censorship, one of the main purposes of this article. As expected, as censorship increases, the frequency of selection decreases for the nonzero coefficients. The point here is to reveal which methods are less affected by this. It can be observed in Figure 6 that the MCL and ENL methods are less affected by censorship in terms of the frequency of selection of nonzero coefficients. However, since these methods are less efficient than the SL, LL, and aLL methods in determining ineffective coefficients, their overall performance is lower (see Table 4 and Table 5). On the other hand, the SL, LL, and aLL methods make a balanced selection for both subsets (no effect-non-zero), which indirectly makes them more resistant to censorship.

Figure 7 presents the different configurations for Scenario 2. It shows both effects of sample size and censoring level increment and bar plots for k=40. The detection performance of the methods of nonzero coefficients is less affected than in Figure 6. However, the selection of the ineffective set plays a decisive role in terms of the performance of each of the methods. For example, MCL and ENL performed poorly in the correct determination of ineffective coefficients when the censorship level increased, but LL, aLL, and SL were able to make the right choice under heavy censorship.

### 5.2. Analysis of Nonparametric Component

This section is prepared to show the behaviors of the estimation of nonparametric components by the introduced six estimators. Performance scores of the methods are given in Table 6 and Table 7, and MSE and ReMSE metrics are used. Additionally, Figure 8 and Figure 9 are provided to show the real smooth function versus all estimated curves for individual simulation repeats. These figures can provide information about the variation of the estimates according to both scenarios and censoring effects. Finally, in Figure 10, estimated curves obtained from all methods are inspected with four different configurations.

Table 6 and Table 7 include the MSE and ReMSE values for the two scenarios. For Scenario 1, the aLL method gives more dominant values than others, followed by SL and LL. As expected, RL shows the worst performance; however, the difference from the others is small. Note that, when the sample size becomes larger, all methods begin to give similar results. Dependent on this similarity, the ReMSE scores become closer to one, which is an expected result. Thus, even if the censoring level increases, ReMSE scores may decrease. If the tables are inspected carefully, as mentioned in Section 5.1, aLL overcomes the censorship problem better than the others regarding Scenario 1, which means that contributions of covariates are high. However, in Scenario 2, SL shows better performance in high censoring levels, especially in small and medium sample sizes. Additionally, it is clearly observed that the number of covariates (k) affects the performances. In Table 7, when k=15, the LL and SL methods show good performances.

Figure 8 shows two different simulation configurations for Scenario 1. The purpose of this figure is to illustrate the effect of censorship in curve estimation. Therefore, panel (i) is obtained for 10% censorship, and panel (ii) for 30% censorship. As can be seen at a glance, the minimum and maximum points of the prediction points obtained from all simulations around the real curve are shown with vertical lines. This reveals the range of variation of the estimators. Accordingly, when the difference between the effect of censorship panel (i) and panel (ii) is examined, it can be seen how the range of variation widens. It can be said, with the help of the values in Table 6, that the estimator with the least expansion is aLL and the method with the most is RL. It should also be noted that the SL and LL methods also showed satisfactory results.

Figure 9 shows the effect of censorship on the estimated curves for Scenario 2 with a large sample size and relatively few covariates (k=15). Because there are too many data points, the lines appear as a black area. Compared with Figure 8, the effect of censorship is less, and the estimators obtain curves closer to the true curve. In addition, due to the large sample size, each method estimated very close curves. This can be clearly seen in Table 7. The obtained performance values were very close to each other. It can therefore be said that the introduced six estimators produce satisfactory results in high samples, and they are relatively less affected by censorship in this scenario.

Figure 10 consists of four panels (a)–(d) containing four different simulation cases. The first two panels (a and b) show the estimated curves of Scenario 1 for different sample sizes, different censorship levels, and different numbers of explanatory variables. It can be clearly seen that the curves in panel (a) are smoother than in panel (b). This can be explained by the messy scattering of synthetic data, which can be observed in all panels. The censorship level increases the corruption of the data structure. Similarly, panels (c) and (d) are obtained for Scenario 2, but only to observe the effect of the change in censorship level. However, the effect of the large sample size is clearly visible, and the curves appear smooth in panel (d), despite the deterioration in the data structure. If examined carefully, the aLL method gives the closest curve to the true curve. At the same time, the other methods have shown satisfactory results in representing the data.

## 6. Hepatocellular Carcinoma Dataset

This section contains the estimation of a right-censored partially linear model for real data, the Hepatocellular carcinoma dataset, by the introduced six estimators (RL, LL, aLL, SL, MCL, and ENL). Their performances are compared, and the results are presented in Table 8 and Figure 11, Figure 12, Figure 13 and Figure 14. The dataset was collected by [31] to study CXCL17 gene expression for hepatocellular carcinoma.

The aforementioned dataset involves 227 data points and 13 explanatory variables, including age, recurrence-free survival ($RFS_{i}$), gender ($Gen_{i}$), and HBsAg ($surface\;antigen\;of\;the\;hepatitis\;B\;virus-HB_{i}$). Some variables that were obtained from blood tests to measure liver damage include $ALT_{i}$ (alanine aminotransferase), $AST_{i}$ (aspartate aminotransferase), and $AFP_{i}\;\left(apha-fetoprotein\right)$. The covariates of tumors detected in the liver are tumor size ($TS_{i}$), $TNM_{i}$ (tumor node and metastasis), $BCLC_{i}\;\left(Barcelona\;Clinic\;Liver\;Cancer\;Staging\;System\right)$ and values of genes related to liver cancer: CXCL17T ($CXCT_{i}$), CXCL17P ($CXCP_{i}$), and CXCL17N ($CXCN_{i}$). Note that the logarithm of the overall survival time ($OS_{i}$) is used as a response variable. Note also that the age variable is used as a nonparametric covariate because of its nonlinear structure. The remaining 12 explanatory variables are added to the parametric component of the model. Accordingly, the right-censored partially linear model can be written as follows:(34)$$log\left(OS_{i}\right)=X_{i}^{T}\beta +f\left(age_{i}\right)+\varepsilon_{i},1\leq i\leq 227$$
where
$$X_{i}^{T}=\left[RFS_{i},\;Gen_{i},\;HB_{i},\;ALT_{i},\;AST_{i},\;AFP_{i},\;TS_{i},\;TNM_{i},\;BCLC_{i},\;CXCT_{i},\;CXCP_{i},CXCN_{i}\right]$$
is the (228×12)-dimensional covariate matrix for the parametric component of the model, and $\beta ={\left(\beta_{1},\ldots ,\beta_{12}\right)}^{T}$ is the (12×1)-dimensional vector of the regression coefficients to be estimated. Note that in the estimation process, $log\left(OS_{i}\right)$ cannot be used directly because of censoring. Therefore, synthetic data transformation is applied to $log\left(OS_{i}\right)$ as in (6). Note also that the dataset includes 84 right-censored survival times, which means that the censoring level is CL=37%. This ratio can be interpreted as a heavy censoring level in the simulation study. Therefore, it is expected that the results of the real data example should be in harmony with the results of corresponding simulation configurations (n=150, 300, k=15, CL=30%).

To describe the hepatocellular carcinoma dataset, Figure 11 and Figure 12 are provided. Figure 11 is constructed by two panels, (a) and (b). In panel (a), a scatter plot of the data points can be seen with censored and noncensored points. As can be observed, there are a lot of right-censored points in the dataset. To solve this problem, synthetic data transformation is realized and is shown in panel (b). The synthetic data give zero value to right-censored points and increase the magnitude of the remaining data. Thus, it aims to make equal the expected values of $Y_{i\hat{G}}$ and completely observed response Y (but we do not know in real cases). Figure 12 presents the plot for response variable log(OS) versus nonparametric covariate age to show the nonlinear relationship between them. Accordingly, a hypothetical curve is presented, which proves our claim on the nonlinear relationship.

General outcomes for the analysis of the hepatocellular carcinoma dataset are presented in Table 8, which involves the performance scores of the six estimators. Note that, here, G-score cannot be calculated due to real regression coefficients being unknown. In Table 8, RL gives the highest $R^{2}$ value because it uses all 12 covariates in model estimation, and sparse and nonzero subsets are considered. The aLL and SL methods provide satisfying values with fewer covariates, especially aLL. Regarding the estimation of the nonparametric component, SL gives the best estimation, which supports our inference given before. In addition, aLL gives a smaller MSE value than the other four estimators.

The estimated coefficients are shown with bar plots in Figure 13 to illustrate how the methods work and to make a healthy comparison. In panels (b) and (c), the similar process of aLL and SL can be observed clearly. The ENL and RL methods also look similar to each other, which can be understood from Table 8.

Figure 14 involves the six fitted curves obtained by the introduced estimators. At first glance, all the fitted curves are very close to each other, which can be monitored in the MSE scores given in Table 8. However, the difference between RL and the other five methods can be easily observed. Due to the data structure having excessive variation, in the modeling process, the local polynomial method gains importance because it takes into account the local densities. This case can be counted as one of the important contributions of this paper.

## 7. Conclusions

The results of the paper obtained from the simulation study are given in Table 2, Table 3, Table 4, Table 5, Table 6 and Table 7 and Figure 4, Figure 5, Figure 6, Figure 7, Figure 8, Figure 9 and Figure 10. The analysis is made for both parametric and nonparametric components of the model individually. The advantage of the simulation study is knowing the real values of the regression coefficients; the accuracy and sensitivity of the estimators are thus evaluated, using the confusion matrix in Table 1. From the results, the aLL and SL estimators showed the best performance and gave satisfactory results for the model estimation. In addition, the behaviors of the methods are inspected for three cases, which are sample sizes (n=50, 150, 300), number of covariates (k=1540), and censoring level (CL=10%, 30%). These effects are also observed by the figures. A real data example using the hepatocellular carcinoma dataset is analyzed using the introduced estimators. The results of that dataset are compared with the related simulation configurations. By using the mentioned results, concluding remarks are given as follows:
From the simulation results regarding the parametric component estimation, Table 2, Table 3, Table 4 and Table 5 prove that the aLL and SL methods give satisfying results in terms of the metrics RMSE, $R^{2}$, accuracy, and G-score. In more detail, for small sample sizes and low censoring levels, SL generally shows better performance than the other five methods. However, for the problematic scenarios, the aLL estimator is the best in both estimation performance and making a true selection between zero and nonzero subsets. Figure 4 and Figure 5 support these inferences.In addition to introduced evaluation metrics, the selection frequency of the estimators is inspected for the simulation study, and results are shown in bar plots given in Figure 6 and Figure 7. These figures demonstrate the consistency of the estimators in terms of their selection of the sparse and nonzero subsets for each coefficient. Under heavy censorship, it can be seen that LL, aLL, and SL gave the best performances. The ENL and MCL estimators did not show a good performance in this case.The introduced six estimators provide closer performances on the estimation of the nonparametric component. Corresponding results are given in Table 6 and Table 7 and Figure 8, Figure 9 and Figure 10. Note that Figure 8 and Figure 9 are drawn to show the individual fitted curves obtained from each simulation, which provides information about the variation of the estimators. Although the estimators give similar evaluation scores and closer fitted curves (which is seen in Figure 10), aLL and SL are the best.In the hepatocellular carcinoma dataset analysis, the outcomes are found in harmony with the corresponding simulation scenarios. The results are provided in Table 8 and Figure 13 and Figure 14. Similar to the simulation study, SL and aLL show the best performance. However, from Figure 14, it can be seen that the fitted curves are very close to each other, which can be explained by the large sample size. The real data study demonstrates that all six estimators show considerably good model estimates, which makes valuable the contribution of the paper.


Finally, from the results of both the simulation and real data studies, the introduced six estimators for the right-censored partially linear models based on penalty and shrinkage strategies are compared, and results are presented. It is found that the adaptive lasso (aLL) and SCAD (SL) methods are more resistant than the other four estimators against the effects of censorship and the number of covariates. In general, the ridge (RL) estimator showed poor performance. On the other hand, the lasso (LL), MCP (MCL), and elasticnet (ENL) methods provided good performance for both the parametric and nonparametric components. This study recommends the aLL and SL estimators for the problematic scenarios.

## Figures and Tables

**Figure 1 entropy-24-01833-f001:**
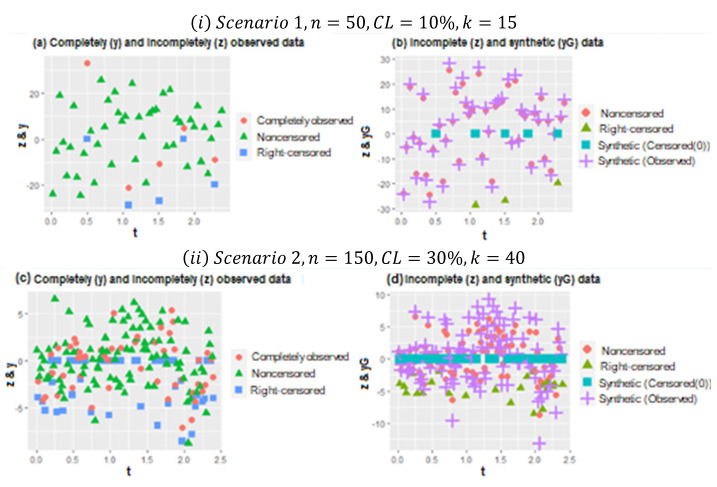
Overview for the generated data based on different simulation configurations.

**Figure 2 entropy-24-01833-f002:**
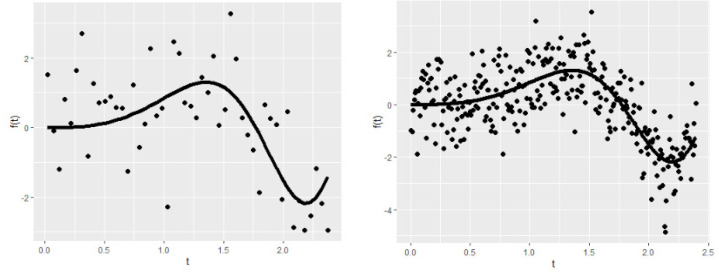
The generated nonparametric component of the semiparametric model for two simulation combinations.

**Figure 3 entropy-24-01833-f003:**
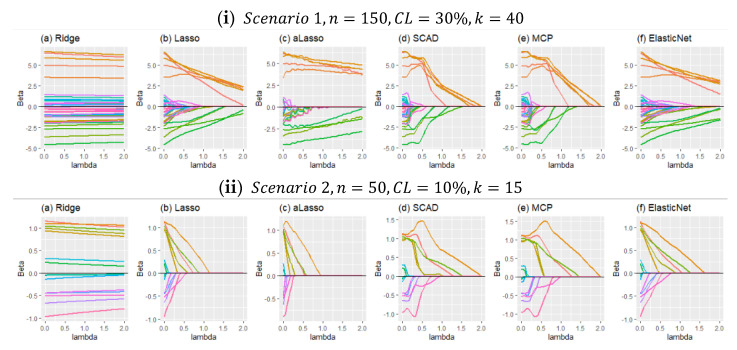
The behaviors of the introduced modified penalty functions according to shrinkage parameters for two different cases.

**Figure 4 entropy-24-01833-f004:**
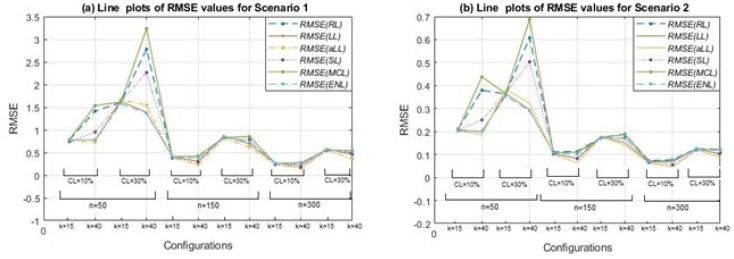
Line plots of RMSE scores given in Table 2 and Table 3 for all simulation configurations.

**Figure 5 entropy-24-01833-f005:**
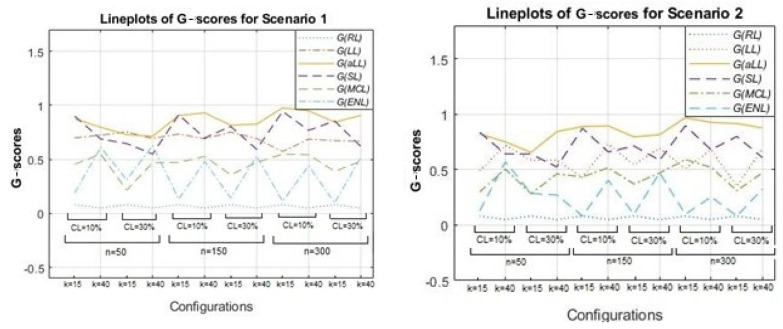
Line plots for G-scores given in Table 4 and Table 5 for all simulation configurations.

**Figure 6 entropy-24-01833-f006:**
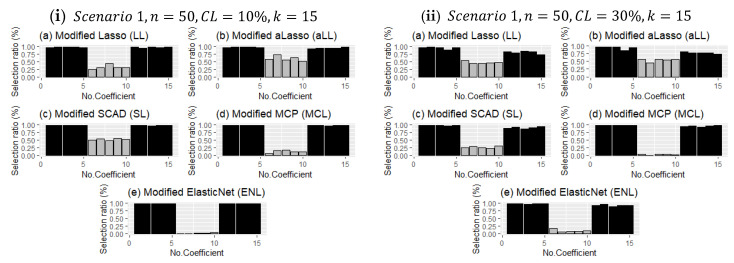
Comparison of the modified estimators regarding deciding the true coefficient for all simulation runs. The *dark-colored* bars denote the selection ratios of *nonzero regression coefficients*, and the *gray-colored* ones represent the ratios for *sparse coefficients*.

**Figure 7 entropy-24-01833-f007:**
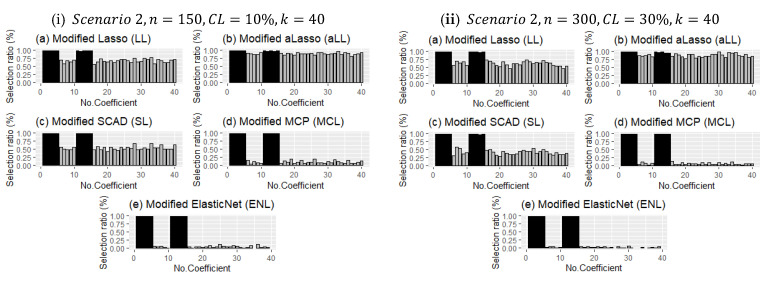
Comparison of the modified estimators as in Figure 6 but for different simulation configurations when k=40.

**Figure 8 entropy-24-01833-f008:**
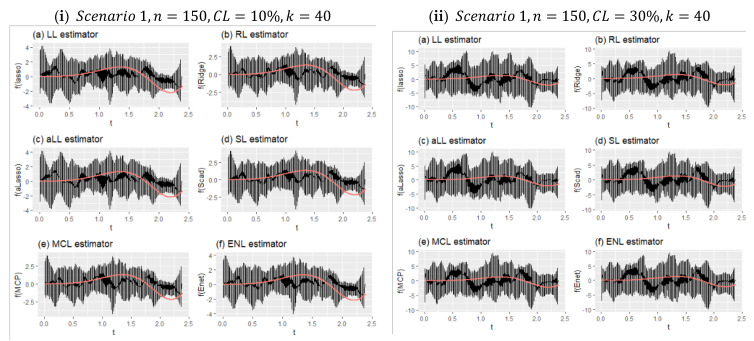
Obtained fitted curves for the methods from all simulation runs for Scenario 1.

**Figure 9 entropy-24-01833-f009:**
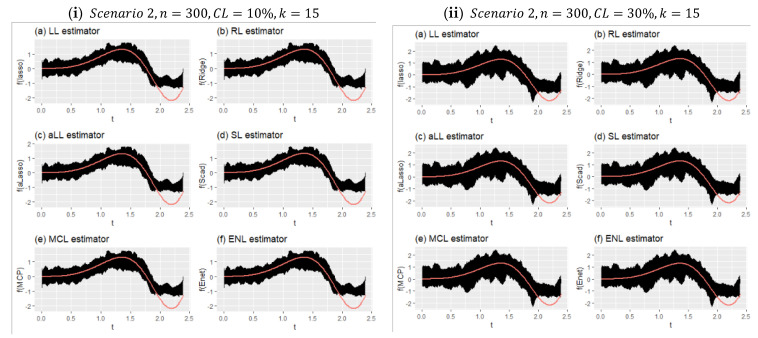
Fitted curves for Scenario 2 for the simulation settings given in (**i**) and (**ii**).

**Figure 10 entropy-24-01833-f010:**
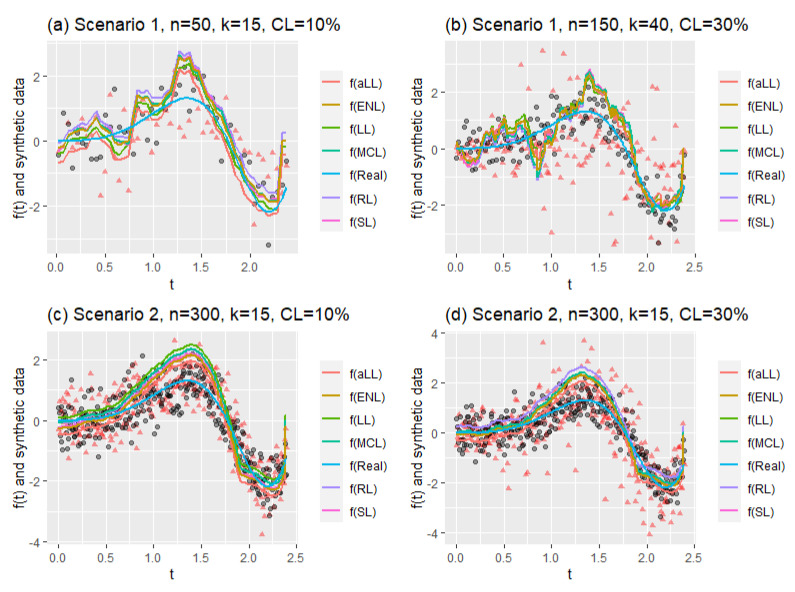
Mean of fitted curves obtained for the different simulation configurations. Red triangles (△) show the synthetic response values (t vs. $Y_{\hat{G}}-X\hat{\beta }$), and black dots (.) denote the original data points t vs. f(t).

**Figure 11 entropy-24-01833-f011:**
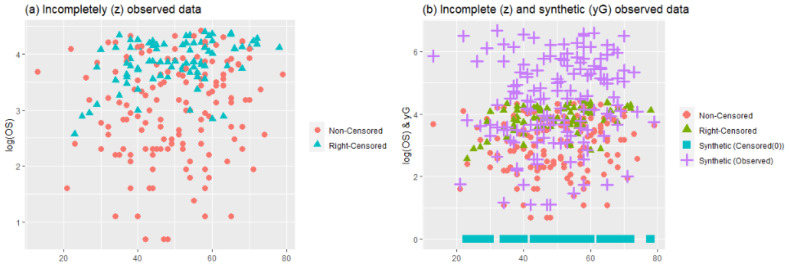
Descriptive plots for the right-censored hepatocellular carcinoma dataset.

**Figure 12 entropy-24-01833-f012:**
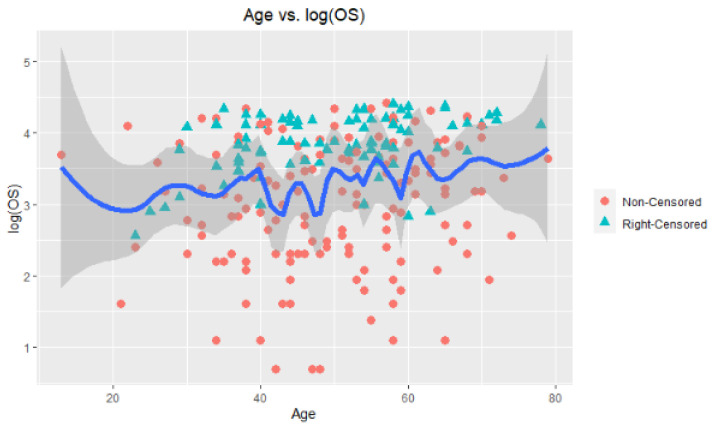
Plot for nonparametric covariate (age) with hypothetical curve (blue line) with confidence intervals (gray areas).

**Figure 13 entropy-24-01833-f013:**
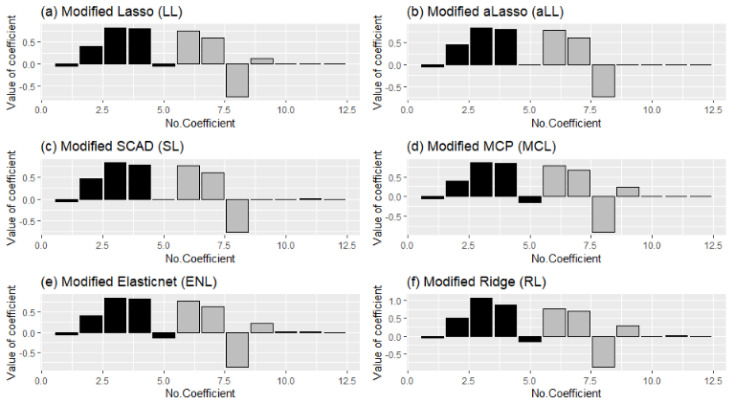
Bar plots of estimated coefficients obtained from the modified six estimators.

**Figure 14 entropy-24-01833-f014:**
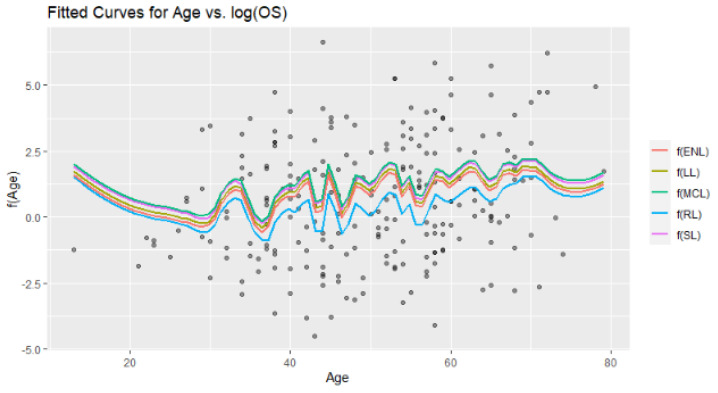
Fitted curves for a nonparametric component of the model obtained from the six modified estimators.

**Table 1 entropy-24-01833-t001:** Confusion matrix for variable selection.

	Covariate of Interest	Irrelevant Covariate
Covariate of interest	*a:* no. true selection of covariate of interest	*b*: no. false selection of covariate of interest
Irrelevant covariate	*c*: no. false selection of irrelevant covariate	*d*: no. true selection of irrelevant covariate

**Table 2 entropy-24-01833-t002:** RMSE and $R^{2}$ values obtained from all simulation runs (Scenario 1).

n	CL	k	${\hat{\beta }}_{RL}$		${\hat{\beta }}_{LL}$		${\hat{\beta }}_{aLL}$		${\hat{\beta }}_{SL}$		${\hat{\beta }}_{MCL}$		${\hat{\beta }}_{ENL}$	
			RMSE	$R^{2}$	RMSE	$R^{2}$	RMSE	$R^{2}$	RMSE	$R^{2}$	RMSE	$R^{2}$	RMSE	$R^{2}$
50	0.1	15	0.779	0.908	0.782	0.905	0.758 *	0.906	**0.743**	**0.909**	0.787	0.908	0.777	0.907
		40	1.414	0.879	0.777 *	0.875	**0.730**	**0.884**	0.952	0.880	1.536	0.876	0.784	0.881
	0.3	15	1.602 *	**0.653**	1.630	0.646	1.670	0.640	1.620	0.651	1.617	0.651	**1.581**	**0.653**
		40	2.779	0.617	1.387 *	0.610	1.547	0.590	2.264	0.618	3.234	0.608	**1.367**	**0.625**
150	0.1	15	0.410	0.949	0.410	0.948	0.386 *	0.948	**0.381**	**0.949**	0.409	0.949	0.411	0.948
		40	0.423	0.942	0.312	0.945	**0.236**	0.950	0.294 *	0.950	0.423	0.942	0.379	0.944
	0.3	15	0.830	**0.811**	0.864	0.805	**0.814**	0.810	0.822 *	**0.811**	0.830	0.810	0.834	0.809
		40	0.854	0.780	0.692 *	0.787	**0.621**	0.799	0.783	0.789	0.855	0.778	0.741	0.791
300	0.1	15	0.263	0.968	0.267	0.967	**0.237**	0.968	0.239 *	**0.969**	0.261	0.968	0.262	0.968
		40	0.276	0.965	0.213	0.968	**0.154**	0.971	0.177 *	0.970	0.273	0.965	0.258	0.966
	0.3	15	0.562	0.895	0.583	0.893	0.550 *	0.895	**0.545**	**0.896**	0.562	0.895	0.568	0.895
		40	0.545	0.878	0.469 *	0.884	**0.363**	**0.897**	0.464	0.889	0.545	0.878	0.512	0.882

**Bold** ones are the best performance scores; *****: the second-best score in RMSE scores.

**Table 3 entropy-24-01833-t003:** RMSE and $R^{2}$ values obtained for Scenario 2.

n	CL	k	${\hat{\beta }}_{RL}$		${\hat{\beta }}_{LL}$		${\hat{\beta }}_{aLL}$		${\hat{\beta }}_{SL}$		${\hat{\beta }}_{MCL}$		${\hat{\beta }}_{ENL}$	
			RMSE	$R^{2}$	RMSE	$R^{2}$	RMSE	$R^{2}$	RMSE	$R^{2}$	RMSE	$R^{2}$	RMSE	$R^{2}$
50	0.1	15	0.211	**0.890**	0.207	0.884	0.204	0.885	0.205 *	0.889	0.209	0.889	**0.203**	0.889
		40	0.379	**0.890**	**0.198**	0.848	0.185 *	0.865	0.251	0.882	0.437	0.885	0.202	0.874
	0.3	15	0.363 *	**0.635**	0.365	0.617	0.384	0.614	0.365	0.629	0.365	0.630	**0.355**	0.632
		40	0.608	0.581	0.295 *	0.543	0.323	0.542	0.503	0.578	0.688	0.570	**0.290**	**0.582**
150	0.1	15	0.112	**0.892**	0.106	0.891	**0.100**	0.891	0.102 *	**0.892**	0.109	0.891	0.108	0.891
		40	0.114	0.899	0.083 *	0.893	**0.064**	0.899	0.083	**0.900**	0.110	0.896	0.101	0.899
	0.3	15	0.174	**0.765**	0.179	0.757	0.176	0.759	**0.171**	0.764	0.173 *	0.764	0.173 *	0.763
		40	0.189	0.737	0.151 *	0.733	**0.141**	**0.743**	0.174	0.740	0.187	0.734	0.169	**0.743**
300	0.1	15	0.074	0.897	0.069	0.897	**0.064**	0.897	0.066 *	0.897	0.071	0.897	0.071	0.897
		40	0.077	0.897	0.068	0.890	**0.045**	0.895	0.055 *	**0.898**	0.074	0.895	0.073	0.895
	0.3	15	0.125	**0.834**	0.126	0.831	0.121 *	0.832	**0.120**	**0.834**	0.123	0.833	0.124	0.833
		40	0.123	0.811	0.102 *	0.809	**0.087**	**0.818**	0.106	0.816	0.121	0.808	0.117	0.810

**Bold** ones are the best performance scores; *****: the second-best score in RMSE scores.

**Table 4 entropy-24-01833-t004:** G-score and accuracy values obtained for all simulation combinations (Scenario 1).

n	CL	k	${\hat{\beta }}_{RL}$		${\hat{\beta }}_{LL}$		${\hat{\beta }}_{aLL}$		${\hat{\beta }}_{SL}$		${\hat{\beta }}_{MCL}$		${\hat{\beta }}_{ENL}$	
			Acc	G	Acc	G	Acc	G	Acc	G	Acc	G	Acc	G
50	0.1	15	0.667	0.082	0.772	0.699	**0.859**	0.874 *	0.845 *	**0.900**	0.715	0.455	0.679	0.190
		40	0.250	0.050	0.767	0.724 *	**0.828**	**0.797**	0.717	0.688	0.439	0.548	0.576	0.612
	0.3	15	0.667	0.082	0.759 *	**0.753**	**0.761**	0.728 *	0.735	0.646	0.673	0.215	0.687	0.311
		40	0.250	0.050	**0.756**	0.694 *	0.746 *	**0.713**	0.561	0.549	0.338	0.470	0.656	0.628
150	0.1	15	0.667	0.082	0.772	0.734	**0.916**	0.905 *	0.859 *	**0.912**	0.716	0.472	0.676	0.131
		40	0.250	0.050	0.707	0.690	**0.952**	**0.930**	0.722 *	0.698 *	0.377	0.527	0.343	0.481
	0.3	15	0.667	0.082	0.780 *	0.752	**0.843**	**0.815**	0.771	0.805 *	0.692	0.357	0.672	0.143
		40	0.250	0.050	0.712 *	0.692 *	**0.868**	**0.826**	0.536	0.593	0.313	0.472	0.388	0.531
300	0.1	15	0.667	0.082	0.768	0.572	**0.963**	**0.976**	0.911 *	0.943 *	0.725	0.549	0.669	0.112
		40	0.250	0.050	0.700	0.687	**0.967**	**0.948**	0.805 *	0.768 *	0.400	0.543	0.322	0.440
	0.3	15	0.667	0.082	0.768	0.673	**0.887**	**0.844**	0.809 *	0.855 *	0.696	0.388	0.668	0.097
		40	0.250	0.050	0.670 *	0.668 *	**0.936**	**0.906**	0.595	0.621	0.319	0.482	0.339	0.516

**Bold** ones are the best performance scores; *****: the second-best score in G-scores and accuracy.

**Table 5 entropy-24-01833-t005:** G-score and accuracy metrics for all simulation combinations (Scenario 2).

n	CL	k	${\hat{\beta }}_{RL}$		${\hat{\beta }}_{LL}$		${\hat{\beta }}_{aLL}$		${\hat{\beta }}_{SL}$		${\hat{\beta }}_{MCL}$		${\hat{\beta }}_{ENL}$	
			Acc	G	Acc	G	Acc	G	Acc	G	Acc	G	Acc	G
50	0.1	15	0.667	0.082	0.740	0.489	**0.837**	0.823 *	0.809 *	**0.838**	0.692	0.299	0.671	0.127
		40	0.250	0.050	0.785 *	0.721 *	**0.793**	**0.751**	0.668	0.642	0.378	0.505	0.516	0.571
	0.3	15	0.667	0.082	0.705	0.585	0.723 *	**0.656**	**0.729**	0.642 *	0.679	0.283	0.675	0.286
		40	0.250	0.050	**0.752**	0.585 *	0.731*	**0.843**	0.526	0.525	0.326	0.462	0.580	0.272
150	0.1	15	0.667	0.082	0.699	0.433	**0.895**	**0.890**	0.841 *	0.873 *	0.701	0.435	0.667	0.082
		40	0.250	0.050	0.757 *	0.729 *	**0.928**	**0.895**	0.665	0.660	0.343	0.517	0.296	0.404
	0.3	15	0.667	0.082	0.749	0.546	**0.837**	0.796	0.767 *	0.714 *	0.692	0.371	0.668	0.097
		40	0.250	0.050	0.721 *	0.693 *	**0.859**	**0.816**	0.517	0.585	0.307	0.470	0.316	0.472
300	0.1	15	0.667	0.082	0.709	0.512	**0.953**	**0.969**	0.860 *	0.897 *	0.724	0.592	0.668	0.097
		40	0.250	0.050	0.625	0.679 *	**0.951**	**0.927**	0.696 *	0.679 *	0.358	0.521	0.292	0.255
	0.3	15	0.667	0.082	0.709	0.353	**0.892**	**0.916**	0.788 *	0.800 *	0.688	0.311	0.667	0.082
		40	0.250	0.050	0.719 *	0.690 *	**0.912**	**0.877**	0.572	0.609	0.313	0.472	0.277	0.327

**Bold** ones are the best performance scores; *****: the second-best score in G-scores and accuracy.

**Table 6 entropy-24-01833-t006:** Performance scores of fitted curves by the modified estimators for Scenario 1.

n	CL	k	${\hat{f}}_{RL}$		${\hat{f}}_{LL}$		${\hat{f}}_{aLL}$		${\hat{f}}_{SL}$		${\hat{f}}_{MCL}$		${\hat{f}}_{ENL}$	
			MSE	ReMSE	MSE	ReMSE	MSE	ReMSE	MSE	ReMSE	MSE	ReMSE	MSE	ReMSE
50	0.1	15	1.26	1.06	1.21	1.01	1.19	0.98	**1.17**	**0.97**	1.21	1.01	1.18	**0.98**
		40	1.26	0.97	1.50	1.20	**1.14**	**0.84**	1.22	0.94	1.50	1.20	1.20	**0.92**
	0.3	15	4.17	1.00	4.29	1.04	**4.11**	0.98	4.15	1.00	4.13	**0.99**	4.14	**0.99**
		40	4.73	1.12	4.59	1.08	4.17	0.90	**3.50**	**0.77**	5.26	1.26	4.07	0.93
150	0.1	15	0.75	1.12	0.67	0.98	**0.62**	**0.89**	0.68	1.00	0.69	1.02	0.69	1.02
		40	0.60	1.04	0.60	1.04	0.61	**1.04**	**0.54**	**0.91**	0.57	0.97	0.58	1.00
	0.3	15	2.38	1.08	2.22	0.99	**2.19**	0.97	2.20	**0.98**	2.20	0.98	2.20	0.98
		40	1.87	1.04	2.13	1.21	**1.65**	0.88	1.75	**0.96**	1.72	0.94	1.81	1.00
300	0.1	15	0.45	1.07	0.43	**1.01**	**0.41**	0.95	0.43	**1.01**	0.42	0.98	0.42	0.98
		40	0.39	1.07	**0.35**	**0.94**	0.36	0.97	0.36	0.97	0.37	1.01	0.38	1.04
	0.3	15	1.29	1.01	1.28	1.00	**1.24**	**0.97**	1.29	**1.01**	1.28	1.00	1.27	1.00
		40	1.12	1.06	1.14	1.08	**1.01**	**0.93**	1.06	0.99	1.04	0.97	1.06	0.99

**Bold** ones are the best performance scores.

**Table 7 entropy-24-01833-t007:** Performance scores of fitted curves by the modified estimators for Scenario 2.

n	CL	k	${\hat{f}}_{RL}$		${\hat{f}}_{LL}$		${\hat{f}}_{aLL}$		${\hat{f}}_{SL}$		${\hat{f}}_{MCL}$		${\hat{f}}_{ENL}$	
			MSE	ReMSE	MSE	ReMSE	MSE	ReMSE	*MSE*	*ReMSE*	*MSE*	MSE	ReMSE	MSE
50	0.1	15	0.85	1.02	0.83	0.99	0.81	0.95	**0.78**	**0.92**	0.87	1.05	0.88	1.06
		40	0.99	1.12	1.18	1.38	**0.72**	**0.71**	0.74	0.79	1.09	1.26	0.85	0.94
	0.3	15	**2.14**	**0.94**	2.27	1.01	2.30	1.03	2.22	0.99	2.29	1.02	2.25	1.00
		40	2.52	1.13	2.67	1.21	2.08	0.86	**1.98**	**0.84**	2.48	1.11	2.10	0.91
150	0.1	15	0.54	0.98	0.56	1.02	**0.52**	**0.94**	0.57	1.04	0.56	1.02	0.56	1.02
		40	0.56	1.21	0.50	1.06	**0.41**	**0.81**	0.45	0.93	0.48	1.01	0.49	1.03
	0.3	15	1.33	1.06	1.29	1.02	**1.21**	**0.94**	1.26	0.99	1.26	0.99	1.26	0.99
		40	1.14	1.09	1.04	0.98	1.06	0.97	**0.94**	**0.86**	1.09	1.03	1.12	1.07
300	0.1	15	0.52	1.02	**0.50**	**0.98**	0.51	1.00	**0.50**	**0.98**	0.52	1.02	0.51	1.00
		40	0.41	1.05	0.39	0.99	**0.35**	**0.84**	0.37	0.93	0.43	1.11	0.43	1.11
	0.3	15	0.87	1.02	**0.84**	**0.98**	0.85	0.99	0.85	0.99	0.87	1.02	0.86	1.00
		40	0.81	1.03	0.77	0.97	**0.76**	**0.95**	0.80	1.01	0.81	1.03	0.80	1.01

**Bold** ones are the best performance scores.

**Table 8 entropy-24-01833-t008:** Scores of the evaluation metrics obtained from the hepatocellular carcinoma dataset.

	RL	LL	aLL	SL	MCL	ENL	n	CL	k
$R^{2}$	**0.406**	0.356	0.355 *	0.357 *	0.363	0.362	227	37%	12
No. $\left(\hat{\beta }=0\right)$	0	3	5	4	3	1			
No. $\left(\hat{\beta }\neq 0\right)$	12	9	7	8	9	11			
$MSE\left(\hat{f}\right)$	3.562	3.646	3.554 *	**3.541**	3.609	3.613			

**Bold** ones are the best performance scores; *****: the second-best scores.

## Data Availability

The simulation dataset can be regenerated by the codes provided in https://github.com/yilmazersin13?tab=repositories accessed on 31 August 2022. The hepatocellular carcinoma dataset is publicly available in R-package named “asaur”.

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
