# Peer review of "Penalty and Shrinkage Strategies Based on Local Polynomials for Right-Censored Partially Linear Regression"

_entropy, 2022, doi:10.3390/e24121833_

Round 1

Reviewer 1 Report

This paper studies the problem of estimating right-censored semiparametric regression model under six different penalty functions and shrinkage strategies. The theoretical contribution seems solid. However, it contains a lot of confusions that should be addressed:

1.     For the special form of eqn. (1.1), is it possible to distinguish the effects of the non-parametric form f(t_i) and the noise \varepsilon_i?

2.     For eqn. (1.3), why are the values of observations defined in such a special way? What is in the intuition of y_i?

3.     More explanations are necessary about why studying the censorship is interesting.

4.     I believe this manuscript contains a lot of missing symbols such as line 85, line 48. This causes some difficulty to follow the article.

5.     The technical contributions seem solid.

Author Response

Comments of the reviewers are answered point-by-point and given in the attached file. All changes are indicated with the MS Word Track Changes. We hope the revision satisfies the Editor and the reviewer.

Reviewer 2 Report

See the attached report.

Author Response

(The authors gave the same response as above.)

Round 2

Reviewer 2 Report

line 147 - The definition of $t_0$ seems odd to me as it seems to imply that there is no dependence on $i$ but the definition seems to be imply otherwise. Wouldn't it be clearer to denote this by $t_{i0}$?

Author Response

We thank to the reviewer for the valuable suggestion. Accordingly we used  $t_{i0}$ notation and revised the paper.